# Insights into the Middle-Late Miocene palaeoceanographic development of Cyprus (E. Mediterranean) from a new $\delta^{18}O$ and $\delta^{13}C$ stable isotope composite record

5 Torin Cannings<sup>1,2</sup>, Alastair H. F. Robertson<sup>1</sup>, Dick Kroon<sup>1</sup>

Correspondence to: Torin Cannings (torin.cannings@unige.ch)

**Abstract.** The Middle to Late Miocene was a time of significant global climate change. In the eastern Mediterranean region, these climatic changes coincided with important tectonic events, which resulted in changes to the organisation of oceanic gateways, altering oceanic circulation patterns. The Miocene Climatic Optimum is regarded as the most recent CO<sub>2</sub>-driven warming event in Earth's climate history and has been proposed as an analogue for future climate change. We present a c. 12 Ma record of oxygen and carbon stable isotopes from the island of Cyprus to help constrain the nature and extent of Miocene palaeoceanographic changes in the eastern Mediterranean region. Cyprus exposes Neogene deep-sea pelagic sedimentary rocks which are suitable for stable isotope studies. Our composite geochemical record integrates data from the Lower to Upper Miocene succession at Kottaphi Hill along the northern margin of the Troodos ophiolite, with the Upper Miocene succession at Lapatza Hill in the north of Cyprus. Calcareous nannofossil biostratigraphy reveals that the composite record spans the Miocene Climatic Optimum's onset to the beginning of the Messinian Salinity Crisis. The new stable isotopic record reveals a complex interplay between global climate change and regional-to-local tectonic changes. In the earlier part of the record, global climate changes dominated; however, by the end of the Late Miocene, tectonic events culminated in isolation of the Mediterranean basins, resulting in a deviation from global open-ocean trends. Strontium isotope analysis is used primarily to help constrain the age of the Miocene successions sampled and implies changes in the connectivity of the eastern Mediterranean basins during the Late Miocene. The combined data provide a useful reference for oceanographic changes in the eastern Mediterranean basins during the Miocene, compared to the global oceans.

#### 30 1. Introduction

Global climate changes during the Miocene are important analogues for modern climatic changes (You, 2010; Steinthorsdottir et al., 2021; Holbourn et al., 2022), especially because the Miocene epoch (c. 23-5 Ma) was a time of significant climatic and environmental changes. For the global oceans, it is believed that a slight cooling during the Early Miocene was followed by a dramatic (c. 4°C) warming

<sup>&</sup>lt;sup>1</sup>School of GeoSciences, Grant Institute, University of Edinburgh, James Hutton Road, Edinburgh EH9 3FE, UK

<sup>&</sup>lt;sup>2</sup>Present address, Department of Earth Sciences, University of Geneva, Geneva, 1205, Switzerland

<sup>&</sup>lt;sup>8</sup> Deceased

- event at the onset of the Miocene Climatic Optimum (MCO), at *c*. 17.9 Ma (Zachos et al., 2008; Holbourn et al., 2015; Westerhold et al., 2020). High temperatures related to the MCO ended at *c*. 14.7 Ma, during a cooling event known as the Middle Miocene Climate Transition (MMCT) (Kennett et al., 1975; Holbourn et al., 2005; Super et al., 2020). Following the MMCT, a gentle cooling trend continued globally until the end of the Miocene (Zachos et al., 2008; Westerhold et al., 2020). Relatively little is
- known about the effects of the above changes regionally, especially in marginal seas such as the eastern Mediterranean. Together with climatic changes, the eastern Mediterranean Sea underwent tectonic changes on a regional to local scale during the Miocene, including extensive changes in ocean circulation patterns related to closure of several ocean gateways (Hsü et al., 1973a; Rouchy and Caruso, 2006; Robertson et al., 2012; Torfstein and Steinberg, 2020). During the Miocene, the oceanic gateway
- which connected the eastern Mediterranean Sea to the Indian Ocean to the east became greatly restricted and eventually closed completely (Hüsing et al., 2009; Torfstein and Steinberg, 2020). The connection between the Mediterranean Sea and the Atlantic Ocean to the west was lost completely during the Late Miocene (Krijgsman et al., 1999a; Flecker et al., 2015). This resulted in the isolation of the Mediterranean Sea from the global open ocean, leading to the well-known Messinian Salinity Crisis
- (MSC) (Hsü et al., 1973a; Rouchy and Caruso, 2006; Roveri et al., 2014).

  Here, we present new oxygen and carbon stable isotope records from Lower to Upper Miocene deepmarine sediments in Cyprus, which provide an excellent reference for the eastern Mediterranean Sea. This new record was constructed in order to better understand the role and interplay of global climatic and regional to local tectonic events in the eastern Mediterranean region during the Miocene.
- We begin with an overview of the Miocene palaeoceanography and tectonically driven changes in ocean connectivity, together with an explanation of relevant aspects of the Miocene geology of Cyprus.

#### 1.1 Miocene $\delta^{18}$ O and $\delta^{13}$ C

- Oxygen stable isotope records, mainly from deep ocean drilling, are the key to identifying and understanding important Miocene climatic events, notably the Miocene Climatic Optimum (MCO)
- (Zachos et al., 2008; Holbourn et al., 2015; Westerhold et al., 2020). Such events are now recognised in palaeoceanographic records globally using δ<sup>18</sup>O records, and more recently using other methods such as Mg/Ca and organic palaeotemperature proxies (Zachos et al., 2001; Westerhold et al., 2005; Zachos et al., 2008; Holbourn et al., 2015; Modestou et al., 2020; Sosdian et al., 2020; Sosdian and Lear, 2020; Super et al., 2020; Westerhold et al., 2020).
- The onset of the MCO was marked by an abrupt decrease in δ<sup>18</sup>O at c. 16.9 Ma. This falling δ<sup>18</sup>O trend, which is recorded in the global oceans, was associated with c. 4°C increase in global temperature (Zachos et al., 2001; Westerhold et al., 2005; Zachos et al., 2008; Westerhold et al., 2020). The MCO represents a c. 3 Myr greenhouse interval, which began at approximately 16.9 Ma and lasted until c. 14.7 Ma (Zachos et al., 2008; Holbourn et al., 2015; Westerhold et al., 2020). The onset of this warming event is commonly attributed to a dramatic rise in CO<sub>2</sub> levels, from c. 400 ppm to 500 ppm (You et al.,
- event is commonly attributed to a dramatic rise in CO<sub>2</sub> levels, from *c*. 400 ppm to 500 ppm (You et al., 2009; Zhang et al., 2013; Holbourn et al., 2015). The Middle Miocene increase in CO<sub>2</sub> can be explained by rapid eruption of the contemporaneous Columbia River basalts (Hodell et al., 1994; Foster et al., 2012; Reidel, 2015; Kasbohm and Schoene, 2018).

The rapid increase in  $\delta^{13}$ C at c. 16.9 Ma indicates the onset of a positive carbon-isotope excursion known as the Monterey Event, which lasted until c. 13.5 Ma (Vincent and Berger, 1985; Holbourn et al., 2007; Holbourn et al., 2015). The approximate co-occurrence of the MCO with the Monterey Event indicates that the warming was coupled with a significant disruption of the global carbon cycle (Vincent and Berger, 1985; Holbourn et al., 2007; Sosdian et al., 2020).

- The increase in organic carbon burial related to the Monterey Event is believed to have driven CO<sub>2</sub> drawdown (Vincent and Berger, 1985; Flower and Kennett, 1993; Flower and Kennett, 1994; Sosdian et al., 2020), ending the MCO and starting the cooling event known as the Middle Miocene Climate Transition. Incorporation of CO<sub>2</sub> into organic material, particularly at continental margins (e.g. Monterey Formation, California, USA) (Pearson and Palmer, 2000), is believed to have resulted in a global decrease in CO<sub>2</sub> levels, following the highs of the MCO. Resultant increases in Antarctic bottom
- water and North Atlantic deep-water production, combined with a reduction in saline water movement from the Indian Ocean to the Southern Ocean, are believed to have resulted in global-scale changes in ocean circulation that, in turn, triggered further cooling (Flower and Kennett, 1994). The cooling event following the MCO is known as the Middle Miocene Climate Transition MMCT. According to δ<sup>18</sup>O-based sea-level reconstructions, the MMCT represents a dramatic three-step global cooling event, which involved three corresponding steps of extreme sea-level fall. The final temperature decrease, and
- corresponding sea-level fall are believed to relate to the establishment of a permanent East Antarctic Ice Sheet (EAIS) (Kennett et al., 1975; Miller et al., 1991a; Kennett et al., 2004; Holbourn et al., 2005; Miller et al., 2005). This cooling event has also been associated with increased continental runoff and sediment flux, as inferred for the central Mediterranean (Malta) (John et al., 2003).
- Following the MMCT, foraminiferal  $\delta^{18}$ O records indicate that global temperatures continued to cool gradually, with temperature and sea level fluctuations on the 1.2 Myr obliquity cycle (Miller et al., 2005; De Vleeschouwer et al., 2017). However, organic palaeothermometry (TEX<sub>86</sub>) of samples from northern Italy (Monte dei Corvi) implies that the western Mediterranean Sea remained warm at the start of the Late Miocene and then cooled dramatically at c. 8 Ma (Tzanova et al., 2015).
- The Late Miocene Carbon Isotope Shift (LMCIS) is equivalent to a c. 1 ‰ decrease in benthic foraminiferal δ<sup>13</sup>C from c. 7.2 to c. 7 Ma, probably due to a decrease in marine organic carbon burial rates (Keigwin, 1979; Keigwin and Shackleton, 1980; Holbourn et al., 2018). The timing of the LMCIS coincides with the onset of the Late Miocene-Early Pliocene Biogenic Bloom (Dickens and Owen, 1999; Diester-Haass et al., 2004; Diester-Haass et al., 2005).
- Eustatic sea-level fall was previously inferred to be a cause of the loss of connection between the Atlantic Ocean and the proto-Mediterranean Sea, culminating in the Messinian Salinity Crisis (Hsü et al., 1973a; Hsü et al., 1973b; ; Adams et al., 1977; Hsü, 1978). However, benthic foraminiferal δ<sup>18</sup>O records now imply that sea level was relatively stable during the Messinian (Miller et al., 2020). Accordingly, it is now widely believed that regional, to local, tectonics, rather than eustasy, were
   critical to the Messinian isolation of the proto-Mediterranean Sea (Flecker et al., 2015).

#### 1.2 Regional tectonics and ocean gateways

Tectonic processes played an important role in the development of the eastern Mediterranean during the Miocene. During the Paleogene, the Southern Neotethys remained partly open, with the first geological

- evidence of incipient continental collision of the Eurasian and North African (Arabian) plates during the Eocene (Robertson et al., 2016; Darin and Umhoefer, 2022). The initial stage of collision resulted in narrowing of the marine connection between the Southern Neotethys and the Indian Ocean by the Late Eocene (Robertson and Parlak, 2024). During the Early Miocene, the collision intensified, greatly reducing the deep-water connection with the Indian Ocean.

  Marine connection between the proto-Mediterranean Sea and the Indian Ocean persisted until the
- Middle to Late Miocene, as indicated by combined palaeontological (Harzhauser et al., 2007), geological (Hüsing et al., 2009) and geochemical (Kocsis et al., 2008; Bialik et al., 2019) data. The remaining limited connectivity still allowed warm surface water to flow from the Indian Ocean through the Mediterranean into the Atlantic Ocean (Bialik et al., 2019) but this terminated during the Late Miocene (c. 11 Ma) (Hüsing et al., 2009) in response to structural tightening of the suture zone.
- During the Messinian, tectonic uplift related to the north westward migration of the African plate resulted in the closure of the Betic Straight and the Riffian Corridors (Weijermars, 1988; Garcés et al., 1998; Krijgsman et al., 1999a; Krijgsman et al., 1999b; Ng et al., 2021). The closure of these ocean gateways eventually eliminated the Mediterranean-Atlantic marine connection, as indicated by sedimentary, magnetostratigraphic and biostratigraphic evidence (Krijgsman et al., 1999a; Flecker et al., 2015).
  - The overall result of the convergence in both the west and the east was that the Mediterranean Sea became isolated from the global ocean at *c*. 6 Ma, triggering the Messinian Salinity Crisis (MSC). During the MSC, in some areas the Mediterranean, sea-level dropped more than 1000m relative to the open ocean (Hsü et al., 1973a; Rouchy and Caruso, 2006), and this coupled with hyper-aridity resulted in the deposition of thick evaporites within the Mediterranean basins after *c*. 5.57 Ma.

# 1.3 Miocene of Cyprus

- The island of Cyprus constitutes three distinct tectono-stratigraphic entities (Fig. 1), each of which has a unique geological history: the Kyrenia Range in the north (Robertson and Woodcock, 1986; Robertson et al., 2024), the Troodos ophiolite (Gass and Masson-Smith, 1963; Moores and Vine, 1971) in the island's centre and the Mamonia Complex in the west (Lapierre, 1972; Robertson and Woodcock, 1979; Robertson, 1990; Robertson and Xenophontos, 1993) (Fig. 1).
  - During the Mesozoic, the Mamonia Complex and the Kyrenia Range formed parts of the continental margin of the Southern Neotethys. During the Late Cretaceous, the Troodos ophiolite formed within the
- Southern Neotethys by spreading above a subduction zone. After tectonic amalgamation during the latest Cretaceous, the Troodos ophiolite and the Mamonia Complex were transgressed by a latest Cretaceous to Oligocene sequence of deep-water pelagic carbonates, known as the Lefkara Formation (Mantis, 1970; Robertson and Hudson, 1973; Robertson, 1976; Kähler and Stow, 1998; Lord et al., 2000, Balmer, 2024). The Troodos ophiolite and the Mamonia Complex were significantly uplifted
- during Oligocene-Early Miocene time. The main drivers were the continental collision between Arabia and the Taurides to the east, and northward subduction/underthrusting of Southern Neotethyan remnants beneath Cyprus (Robertson, 1998). The uplift resulted in relative sea-level fall, allowing the deposition of variable, shelf-depth sediments of the Miocene Pakhna Formation in southern Cyprus; these include

hemipelagic carbonates, gravity-flow deposits and localised coral reefs (Robertson, 1977; Follows and Robertson, 1990; Follows et al., 1996; Cannings et al., 2021).

The Pakhna Formation encompasses, two reef-related members. The stratigraphically lower of these units, the Lower Miocene Terra Member is dominated by framestones, with high levels of algal and coral biodiversity (Follows et al., 1996; Coletti et al., 2021). The stratigraphically higher Upper Miocene Koronia Member comprises a reef facies with abundant poritid corals (Follows and Robertson, 1990; Coletti et al., 2021), together with an off-reef facies composed of rhodolith-rich packstones, calcarenites and chalks with scattered bivalve and poritid fragments (Follows and Robertson, 1990). Of particular relevance here, the Pakhna Formation includes nannofossil ooze (chalk) and planktic foraminifer-rich calcareous mudstones (marls) which were sampled for this study on the northern margin of the Troodos ophiolite.

The Pakhna Formation is overlain by various gypsum facies known as the Kalavasos Formation, that are associated with the onset of the MSC (Hsü et al., 1973b; Krijgsman et al., 2002; Wade and Bown, 2006; Manzi et al., 2016). The gypsum accumulated in several small tectonically controlled basins around the periphery of the Troodos Massif (Robertson et al., 1995).

The Miocene sedimentary succession of the Kyrenia Range in the north of Cyprus differs strongly from that of southern Cyprus. The Miocene of the north of Cyprus encompasses an Upper Eocene to Upper Miocene succession of variably deformed non-marine to relatively deep-marine conglomerates. sandstones and mudstones, known as the Kythrea (Değirmenlik) Group (McCay et al., 2013; Robertson et al., 2014; Chen et al., 2022). The Early to Middle Miocene interval is dominated by mudrocks and siliciclastic gravity-flow deposits, which were deposited in a trench or foreland basin setting along the 175 northern margin of the southern Neotethys (McCay et al., 2013). By the Late Miocene, the input of sand-sized siliciclastic sediments waned, allowing Tortonian mudstones and marls, known as the Yılmazköy Formation, to accumulate. With a further decrease in siliciclastic input, dominantly hemipelagic marls, associated with thin (

Figure 1 Outline geological map of Cyprus showing the Troodos Massif, the Mamonia Complex and the Kyrenia Range, as well as their sedimentary cover (modified from Kinnaird et al., 2011). Site locations for this study are also shown. Inset: tectonic setting of Cyprus in the eastern Mediterranean region (modified from Follows and Robertson, 1990; Robertson et al., 1991 and Payne and Robertson, 1995).

#### 2 Methods

#### 2.1 Site selection

interval (c. 17.5 – 5.33 Ma), several exposures were selected for logging and sampling. The approximate ages were already known for previously studied exposures. However, the likely age ranges of the new sections needed to be inferred initially based mainly on lithological correlations. The samples from the new successions were dated during this work using calcareous nannofossil biostratigraphy (Cannings, 2024). Following fieldwork and preliminary dating, two sections were selected with the aim of producing a composite succession spanning c. 17.5 – 5.33 Ma, one on the north side of the Troodos ophiolite at Kottaphi Hill, and the other to the south of Kyrenia Range at Lapatza Hill (Fig. 1). Other sections that proved to be less suitable for the composite succession are detailed in Cannings (2024).

#### 2.1.1 Kottaphi Hill

- Kottaphi Hill (Κοτάφι, Kotaphi or Kottafi) (35° 2'50.40" N 33° 9'13.60" E) is located to the north of Agrokipia (Αγροκηπιά) village (Nicosia District) on the northern margin of the Troodos ophiolite (Fig. 1). Kottaphi Hill is a classic Miocene section of the Pakhna Formation (Mantis, 1972) and that has been the subject of several previous studies (Follows and Robertson, 1990; Davies, 2001; Penttila, 2014; Athanasiou et al., 2021; Coletti et al., 2021). The Miocene succession (Fig. 2a & b) begins with c. 60m of cyclically bedded chalk and marl (i.e. couplets), representing the lower part of the Pakhna Formation (undifferentiated). This pelagic-hemiplegic interval is abruptly overlain by the Koronia Member, an interval of debris-flow deposits that are dominated by clasts (mainly angular) of shallow-water
- (undifferentiated). This pelagic-hemiplegic interval is abruptly overlain by the Koronia Member, an interval of debris-flow deposits that are dominated by clasts (mainly angular) of shallow-water carbonates, including poritid corals (Follows and Robertson, 1990; Follows, 1992).

  Previously, Davies (2001) logged and sampled the Kottaphi Hill succession at a resolution of > 20
- cm), with the main aim of determining the relative roles of climate versus tectonics on sedimentation. The succession was dated as Burdigalian-Tortonian using calcareous nannofossil biostratigraphy, supported by limited carbon and oxygen stable isotope data. Davies (2001) described a thin (c. 5cm thick) cemented trace fossil-rich manganese and iron-rich layer in the mid part of the succession, and interpreted this as a hardground. Based on spectral analysis, Davies (2001) suggested that the chalk-
- marl cyclicity (couplets) at Kottaphi Hill was controlled by regional to global changes in climate and sea-level; i.e. Milankovitch cyclicity. Davies (2001) also identified the Monterey Excursion from the available  $\delta^{13}$ C record, and linked a positive  $\delta^{18}$ O trend from 13 to 14 Ma with the isolation of the Mediterranean from the Indian Ocean. Building on Davies's (2001) study, Penttila (2014) used calcareous nannofossil biostratigraphy and strontium isotope dating, together with foraminiferal
- abundance, and selected X-ray diffraction analysis to improve understanding of the Kottaphi Hill succession. She particularly identified an incoming of feldspar and clay-rich lithic fragments at *c*. 10.6 Ma. This suggested that the adjacent Troodos Massif was near or above the seafloor by this time, allowing erosion of extrusive ophiolitic rocks to take place. Penttila (2014) also discovered 'ghost sapropels' (<2% total organic carbon) towards the top of the marl-chalk succession (beneath the
- Koronia Member).
  Subsequently, Athanasiou et al. (2021) collected samples from a 42.1 m-thick interval of marl-chalk on the lower slopes of Kottaphi Hill. The samples collected underwent calcareous nannofossil, pollen and palynomorph, δ<sup>13</sup>C and δ<sup>18</sup>O and total organic carbon analysis. The authors inferred that the organic carbon-rich layers at Kottaphi Hill were deposited in warm oligotrophic seawater, with strong water
- column stratification, and further suggested that these layers represent the precursors to the sapropels. They also identified the Mi3–Mi5 events (Miller et al., 1991a; Miller et al., 2020) and the CM5–CM7 episodes (Woodruff and Savin, 1985, 1991) in their  $\delta^{13}$ C and  $\delta^{18}$ O stable isotope record. Pollen and palynomorph analysis indicated that land nearby was densely vegetated before c. 14.5 Ma. After c. 13 Ma, a more open landscape existed, accompanied by nearby high rates of soil erosion.

# 235 2.1.2 Lapatza Hill

A c. 60m-thick succession of well-bedded, laterally continuous marl that is exposed on Lapatza Hill (Lapatza Vouno) (35°14'44.70" N 33° 8'35.80" E) (Fig. 1, Fig. 2c) belongs to the Tortonian-Lower Messinian Yazılıtepe Formation. The succession includes numerous thin (

Figure 2 Photographs showing the Kottaphi Hill succession. (a) with the first interval of high-resolution sampling indicated (b), and the sampled succession at Lapatza Hill (c), indicating the approximate sampling transect.

#### 2.2 Sampling

Samples were collected by hand using a geological hammer and a chisel to dig out and collect sediments so that they were unaffected by surface processes as far as possible. A shovel was sometimes

needed to excavate a trench and take samples, especially where loose sediment cover needed to be removed, as at Lapatza Hill. The samples were collected at 5-25 cm intervals to facilitate high-resolution geochemical records. The sampling resolution varied for both sections based on the time interval covered and the inferred sedimentation rate. Previous studies (Davies, 2001; Penttila, 2014; Athanasiou et al., 2021) suggested that the section between 10.8 and 29.5 m at Kottaphi Hill coincided with the MCO and MMCT. Therefore, this part of the succession was subjected to the highest-resolution sampling (5cm). Preliminary dating of the Lapatza Hill succession indicated a high (1-3 cm/kyr) sedimentation rate; therefore, a larger (25 cm) sampling interval was used for this section. Sedimentary logs detailing the lithologies, sedimentary structures and macrofossils (where present) were made for both successions during the sample collection.

#### 260 2.3 Foraminifera

The planktic *Praeorbulina-Orbulina* lineage (Blow, 1956; Pearson et al., 1997; Aze et al., 2011; Spezzaferri et al., 2015) was used for the geochemical analysis. The foraminiferal species are present within the target time interval and are not known to vary in vital effects, depth habitat or symbiotic association (Pearson et al., 1997).

Samples were washed using a 32 µm sieve to remove fine clay and silt fractions. Samples which were not easily broken up by wet sieving were soaked in Milli-Q water and placed on a shaker plate for 90 minutes before being washed again over a 32 µm sieve. This process was repeated up to 3 times for the most resistant samples. The foraminiferal samples were then dried in an oven at 30°C and dry sieved into size fractions for foraminiferal picking.

# 270 2.4 Calcareous nannofossil biostratigraphy

Calcareous nannofossil biostratigraphy was used successfully in Cyprus in several previous studies (Morse, 1996; McCay et al., 2013; Robertson et al., 2019; Cannings et al., 2021). Calcareous nannofossils were identified using smear slides prepared using a standard method (Backman and Shackleton, 1983; Bown and Young, 1998) and then examined with a light microscope at a magnification of ×1000–1250. The calcareous nannofossil biostratigraphy used here is based on Backman et al. (2012), and the results presented are consistent with the updated Miocene Mediterranean biostratigraphy of Di Stefano et al. (2023).

#### 2.5 Strontium isotope dating

- The well-established strontium isotope dating technique (Elderfield, 1986) was previously used for the dating of mainly Neogene carbonate sediments in Cyprus (McCay et al., 2013; Penttila, 2014; Cannings et al., 2021). The isolation of the Mediterranean Sea during the Messinian Salinity Crisis (MSC) is believed to have resulted in anomalous <sup>87</sup>Sr/<sup>86</sup>Sr ratios during this time, such that samples of this age range cannot be reliably dated using this method (Flecker and Ellam, 1999; Flecker et al., 2002; Flecker and Ellam, 2006).
- Five samples of planktic foraminifera from the *Praeorbulina-Orbulina* lineage from the Kottaphi Hill succession, and five samples of planktic foraminifera also from the *Praeorbulina-Orbulina* lineage from

the Lapatza Hill succession were dated using the strontium (Sr) isotope method. These planktic foraminiferal samples were then analysed to help test and constrain the dating of the overall composite succession, as achieved by biostratigraphy. The strontium isotope analysis was carried out at the Scottish Universities Environmental Research Centre using a VG-Sector-54 thermal ionization mass spectrometer. Further details of the methodology used for strontium isotope analysis are provided in the

Strontium isotopic ages were calculated using the LOWESS Sr isotope Look-Up Table (Version 4: 08/04) (McArthur et al., 2001; McArthur and Howarth, 2004). The total errors were calculated by combining the uncertainties of the Sr isotopic analyses with the error of the LOWESS Sr isotope Look-Up Table (Version 4: 08/04) (McArthur et al., 2001; McArthur and Howarth, 2004).

#### 2.6 Stable isotope analysis

supplementary material.

A bulk carbonate (fine fraction ≤63 um) and a planktic foraminiferal stable isotope record are presented below for the composite succession that was obtained from the Kottaphi Hill and Lapatza Hill successions. The combination of both bulk carbonate (fine fraction) and planktic foraminiferal records provides the most complete isotope record possible for the target time interval by reducing the gaps in the record to a minimum. However, bulk carbonate and planktic foraminiferal records can differ due to variable vital effects in the biogenic components (Anderson and Cole, 1975; Reghellin et al., 2015); i.e. a planktic foraminiferal sample contains only one biogenic component, whereas a bulk carbonate sample contains multiple biogenic components and also their fine-grained matrix. For the bulk analysis, 640 samples were broken up and dry sieved through a 38 µm sieve in order to collect a concentrated sample of nannofossil carbonate and to reduce the likelihood of terrigenous components being present. The samples were then ground using a porcelain pestle and mortar to ensure a consistent, homogeneous fine powder.  $\sim 0.5$  mg of each powdered sample was then taken for isotopic analyses. All of the stable isotope measurements of the bulk carbonate (fine fraction) samples were made in the Wolfson stable isotope ratio mass spectrometry suite at the University of Edinburgh on an Elementar precision continuous flow stable isotope ratio mass spectrometer, using a Gilson autosampler equipped with an automated acidification system, heated tray and an iso FLOW system. Isotopic measurements of 578 foraminifera samples were made in the Wolfson stable isotope ratio mass spectrometry suite at the School of GeoSciences at the University of Edinburgh using a dual-inlet Thermo Electron Delta + Advantage stable isotope mass spectrometer, interfaced with a Kiel carbonate II device.

#### 2.7 Age model

New age models were generated for both the Kottaphi Hill and Lapatza Hill successions.

For each succession, a preliminary age model was created based on calcareous nannofossil biostratigraphy. Sedimentation rates were calculated based on the depth/height difference between samples identified as bio-horizons. Samples that were not biostratigraphically dated were assigned tentative ages using a linear interpolation based on the calculated sedimentation rate. The final age model was produced using AnalySeries 2.0.8 (Paillard *et al.*, 1996). The new δ<sup>18</sup>O<sub>planktic</sub> record

- produced for this study was used for this purpose and aligned with the North Atlantic  $\delta^{18}$ O<sub>benthic</sub> compilation record of Cramer *et al.* (2009).
  - For the alignment, the new  $\delta^{18}O_{planktic}$  record was filtered using multiple steps (20, 60, 100 ky) of box filtering, in order to reduce 'noise' and reveal only long-term trends and major perturbations. Identifiable marine isotope stages/events and major trends within the record were aligned with the
- reference record by creating tie points at the midpoints of such trends. This age modelling technique has uncertainty resulting from a combination of the uncertainties associated with the calcareous nannofossil biostratigraphy, the  $\delta^{18}O_{planktic}$  record of this study and the  $\delta^{18}O_{benthic}$  record used for the alignment (Cramer *et al.*, 2009).
- Age models were produced separately for both the Kottaphi and Lapatza hill successions using the same method. These records were then compared carefully to identify the most suitable splice point to combine the two records and so produce a composite record for the whole of the target time interval.

#### 3 Results

#### 3.1 Sedimentary logs

#### 3.1.1 Kottaphi Hill

- The sampled Kottaphi Hill succession begins *c*. 10 m stratigraphically below the prominent hard layer which was previously interpreted as a 'hardground' (Davies, 2001). A simplified stratigraphic log of this succession is shown in Figure 3a, while a more detailed log is recorded in the supplementary material.
- The measured succession at Kottaphi Hill begins with 10.97m of alternating beds of light buff-coloured chalk and slightly darker buff-coloured calcareous marl. At 10.97m, there is an abrupt lithological change to three hard grey limestone beds (15-30cm thick), interbedded with darker, buff-coloured micritic limestone. The upper surfaces of each of the three limestone beds are characterized by well-preserved trace fossils (*Thalassinoides*, *Diplocraterion* and *Zoophycos*), with evidence of preferential cementation and metal-oxide precipitation. This interval is here reinterpreted as a firmground rather
- than a hardground (Cannings, 2024). Above the final cemented micritic limestone bed, chalk and calcareous marl interbeds start at 11.3m. These beds are pinkish-red gradually fading to buff coloured over *c*. 4m. Up section, the abundance of chalk beds increases relative to marl beds. From 27.79m to 32.96m, thick (30-85cm) chalky marl beds are interbedded with thinner (5-20cm) marl beds. Several soft marl beds within this facies have a distinctive dark reddish-brown colour. These marl layers are interpreted as oxidised (i.e. "ghost") sapropels. Above this level, exposure is patchy to rare, mostly
- covered by reworked debris from the above units, and so was not sampled for stable isotope analysis.

#### 3.1.2 Lapatza Hill

The succession sampled at Lapatza Hill (Fig. 3b) comprises 63.1m of marl, with infrequent layers of darker brown marl, dark red marl and chalky marl. Fine parallel laminations are present throughout the succession, becoming more prominent towards the top of the succession. Sedimentary structures

(including observable bioturbation) are otherwise absent. This succession is abruptly overlain by a *c*. 15 m-thick interval of fine-grained alabastrine gypsum; this continues to the top of the hill where the section ends.

The marls are very rich in planktic foraminifera at the base; however, at *c.* 52.5m their abundance decreases especially in the softest marl beds. By *c.* 61m, foraminifera are completely absent. However, some foraminifera reappear in chalky marl beds at *c.* 62m although their abundance rapidly decreases towards the contact with the overlying gypsum. Additionally, numerous thin (1-5 cm) dark brown and red marl manganese and iron-rich layers occur within the entire marl succession, although these were not sampled.

**Figure 3** Simplified stratigraphic logs of the Kottaphi Hill succession (a) showing the occurrence of sedimentary structures and fossils up the section and the Lapatza Hill succession (b) showing the approximate abundance of planktic foraminifera and the main lithological changes up the succession.

#### 3.2 Calcareous nannofossil biostratigraphy

# **375 3.2.1 Kottaphi Hill**

Figure 4a shows the ages obtained using calcareous nannofossil biostratigraphy for the samples from the Kottaphi Hill succession. Accordingly, the base of the succession is between 19.01 and 17.96 Ma, as

indicated by the presence of *Sphenolithus belemnos* (Backman et al., 2012). The top of the sampled succession is slightly younger than 8.8 Ma, as indicated by an increase in the relative abundance of *Reticulofenestra pseudoumbilicus* just beneath the top of the succession. These dates indicate that the Kottaphi Hill succession has an average sedimentation rate of *c*. 0.25 cm/ky. Calcareous nannofossil biostratigraphy indicates that the highest sedimentation rate was 0.33 cm/ky between 11.88 Ma and 11.60 Ma, and the lowest was 0.19 cm/ky between 17.75 Ma and 15.73 Ma. These sedimentation rates do not take account of diagenetic compaction that was, however, minor because there is no evidence of deep burial or deformation of either of the two successions studied.

#### 3.2.2 Lapatza Hill

Figure 4b shows the ages inferred using calcareous nannofossil biostratigraphy for the samples from the Lapatza Hill succession. The increase in the relative abundance of *Reticulofenestra pseudoumbilicus* slightly above the base indicates that the base of the succession is older than 8.8 Ma (using the biostratigraphy of Backman *et al.*, 2012). The first occurrence of *Amaurolithus primus* at *c.* 19.5m indicates an age of 7.39 Ma. These dates indicate that the Lapatza Hill succession has an average sedimentation rate (non-decompacted) of *c.* 1.01 cm/ky. The calcareous nannofossil biostratigraphy indicates that the highest sedimentation rate was 1.08 cm/ky between 8.8 Ma and 7.39 Ma, whereas the lowest sedimentation rate was 0.92 cm/ky between 7.39 Ma and 7.09 Ma.

The calcareous nannofossil assemblages above the 25m level in the Lapatza Hill succession are not agediagnostic. Samples from the upper part of the section contain a diverse calcareous nannofossil assemblage; however, age-indicative species are absent. *Discoaster spp.* specimens are abundant in samples from the lower part of the succession but disappear above *c.* 60m.

Figure 4 Stratigraphic logs of the successions sampled at Kottaphi Hill (a) and Lapatza Hill (b) with height and nature of bio-horizons annotated, together with ages according to Backman et al. (2012). Bio-horizons based on relative abundance (Backman et al., 2012) are less quantitative than other bio-horizons and are marked with a dashed line.

#### 3.3 Strontium isotope dating

#### 3.3.1 Kottaphi Hill

Sr isotope dating indicates that the succession at Kottaphi Hill, above the carbonate firmground, ranges from 16.22 Ma (with an error range of 16.01 Ma to 16.46 Ma) to 7.25 Ma (with an error range of 6.86 Ma to 7.82 Ma) (Fig. 5). The Sr isotopic ages retain their depositional order, with an average sedimentation rate (non-decompacted) of 0.27 cm/ky.

Figure 5 Sr age data for planktic foraminiferal samples from Kottaphi Hill plotted against absolute age. The determined age and the total combined error are shown for each sample.

#### 3.3.2 Lapatza Hill

The Sr isotope data for the Lapatza Hill succession indicate an age range of 11.45 Ma (with an error range of 10.9 Ma to 12.24 Ma) to 5.44 Ma (with an error range of 5.139 Ma to 5.66 Ma) (Fig. 6). The oldest age (lowest <sup>87</sup>Sr/<sup>86</sup>Sr) is for sample TC22 330 from the 31.5 m level, whereas younger ages were calculated for the samples both above and below this. Either the heights of the samples up-section do not correlate with their age, or the ages determined do not represent the timing of biomineralization of these samples of planktic foraminifera. However, there is no sedimentological or biostratigraphic evidence of the first alternative. Additionally, sample TC22 470, taken from a height of 62.75m has a strontium isotopic value (0.709241) outwith the LOWESS Sr isotope Look-Up Table (Version 4: 08/04) (McArthur *et al.*, 2001; McArthur and Howarth, 2004), indicating that this does not represent a global seawater composition.

Figure 6 Sr age data for planktic foraminiferal samples from Lapatza Hill plotted against absolute age. The determined age and the total combined error are shown for each sample.

# 3.4 Age model

The new age model for the Kottaphi Hill indicates that the succession sampled spans 18.96 Ma (Burdigalian) to 7.72 Ma (Tortonian). From the base of the succession to 10.97 m (17.15 Ma), the average sedimentation rate (non-decompacted) is calculated as 0.805 cm/ky, with an average (slower) sedimentation rate of 0.245 cm/ky between 10.97 m (17.15 Ma) and the top of the succession sampled. The new age model for Lapatza Hill spans 9.14 Ma (Tortonian) to 5.74 Ma (Messinian). From the base of the succession to 25 m (6.82 Ma) the average sedimentation rate is calculated as 1.099 cm/ky, with little variation. From 25 m (6.82 Ma) to the top of the succession, the sedimentation rate increased (3.627 cm/ky) (again non-decompacted).

The two sampled successions were combined to produce the composite temporal record. Figures showing the age model, together with the points used for its construction and the sedimentation rates derived from the age models for both sections are provided in the supplementary material. The composite temporal record points to any changes in climatic and environmental conditions throughout the whole of the time interval of interest. A small peak of more positive δ<sup>18</sup>O values in the δ<sup>18</sup>O<sub>planktic</sub> record is noted in both successions at 8.60 Ma. This level (8.60 Ma) was selected as the splice point to combine the two records thus create the representative composite record, taking account of the similarity in δ<sup>18</sup>O<sub>planktic</sub>, δ<sup>18</sup>O<sub>fine fraction</sub>, δ<sup>13</sup>C<sub>planktic</sub> and δ<sup>13</sup>C<sub>fine fraction</sub> trends and values at this level. These spliced intervals provide a c. 13.2 My record, from 18.95 Ma (Burdigalian) to 5.74 Ma (Messinian). This composite record covers the whole of the time interval of interest, from the onset of the Miocene Climatic Optimum to the onset of the Messinian Salinity Crisis.

#### 3.5 Stable Isotopes

#### 3.5.1 $\delta^{18}$ O

- Figure 7 shows the  $\delta^{18}O_{planktic}$  and  $\delta^{18}O_{fine\ fraction}$  records for the composite temporal record and allows comparisons between the two methods. Both records show a generally similar trend for the Kottaphi Hill succession. Although the  $\delta^{18}O_{planktic}$  record appears to be more variable than the  $\delta^{18}O_{fine\ fraction}$  record, both display approximately synchronous peaks and troughs. Prior to the beginning of the  $\delta^{18}O_{planktic}$  record, the  $\delta^{18}O_{fine\ fraction}$  record shows a very gradual decrease in  $\delta^{18}O$  (more negative values).
- The two records show a marked divergence at the splice point, at the beginning of the sampled Lapatza Hill succession. While the δ<sup>18</sup>O<sub>planktic</sub> record for Lapatza Hill continues at a similar δ<sup>18</sup>O value to the Kottaphi Hill succession, the δ<sup>18</sup>O<sub>fine fraction</sub> record exhibits a major (c. 6.0 ‰ VDPB) offset between the two successions. The δ<sup>18</sup>O<sub>fine fraction</sub> time series also records increased variability in the δ<sup>18</sup>O values for the Lapatza Hill succession compared to the Kottaphi Hill succession. The two stable isotope records (planktic and fine fraction sample) show different trends for the Lapatza Hill succession. However, both records show the most positive δ<sup>18</sup>O values for the Lapatza Hill succession at c. 6 Ma.

Figure 7 Plot of  $\delta^{18}O_{planktic}$  (orange) and  $\delta^{18}O_{fine}$  fraction (blue) versus age for the composite temporal record. Measurements are shown relative to VDPB. All measured data are shown with a thin line and box markers; bold lines show the  $\delta^{18}O$  data smoothed by a locally weighted function over 60 kyr (see supplementary material for additional information).

#### 465 3.5.2 $\delta^{13}$ C

The  $\delta^{13}$ C data for the composite temporal record are shown in Figure 8. For the Kottaphi Hill succession, the two records (planktic and fine fraction sample) follow a similar trend, although the  $\delta^{13}$ C<sub>planktic</sub> record is more variable than the  $\delta^{13}$ C<sub>fine fraction</sub> record. The  $\delta^{13}$ C<sub>planktic</sub> values record a steep increase to more positive values in  $\delta^{13}$ C at c. 14.6 Ma, as recorded in the  $\delta^{13}$ C<sub>fine fraction</sub> record. Both records show a significant shift towards more negative values at the splice point between the Kottaphi Hill and Lapatza Hill records. In the  $\delta^{13}$ C<sub>planktic</sub> record, shortly above this shift, the values return to a similar level (1-2 ‰), as in the Kottaphi Hill succession. The shift in  $\delta^{13}$ C is more dramatic (c. 4.5 ‰)

in the  $\delta^{13}C_{\text{fine fraction}}$ ; the values do not return to a similar level as in the Kottaphi Hill succession but instead remain lower. Following the splice point, the  $\delta^{13}C_{\text{planktic}}$  and  $\delta^{13}C_{\text{fine fraction}}$  records show discrete trends. The  $\delta^{13}C_{\text{planktic}}$  values record a second shift to more negative  $\delta^{13}C$  values at c. 7.6 Ma, followed by a 'saw-tooth' decrease until c. 6.5 Ma when values sharply increase, before once again decreasing until the end of the record. In the  $\delta^{13}C_{\text{fine fraction}}$  record, the values continue to decrease following the negative shift at the splice point, until c. 8.5 Ma when the values begin to increase before rapidly increasing to c. 0.00 ‰ at c. 7 Ma. The  $\delta^{13}C_{\text{fine fraction}}$  values steadily decrease from c. 6.9 Ma until the end of the record.

Figure 8 Plot of  $\delta^{13}C_{planktic}$  (orange) and  $\delta^{13}C_{fine fraction}$  (blue) versus age for the composite temporal record. Measurements are shown relative to VDPB. All measured data are shown with a thin line and box markers; bold lines show these  $\delta^{13}C$  data smoothed by a locally weighted function over 60 kyr (see supplementary material for additional information).

#### 4. Discussion

# 4.1 Composite Early-Late Miocene temporal record

By combining the Kottaphi Hill and the Lapatza Hill data, a composite temporal record for the late Early Miocene (Burdigalian) to the Late Miocene (Messinian) can be constructed (Fig. 9). As this composite temporal record combines data from the two different successions sampled, specific features of the two sections (e.g. facies; diagenesis) need to be taken into account, as detailed in Cannings (2024).

Figure 9 Stratigraphic logs of both the Kottaphi Hill succession and the Lapatza Hill succession, shown scaled to age, together with the δ<sup>18</sup>O<sub>planktic</sub> and δ<sup>18</sup>O<sub>fine fraction</sub> records for the composite temporal record and the δ<sup>13</sup>C<sub>planktic</sub> and δ<sup>13</sup>C<sub>fine fraction</sub> records for the composite temporal record. All of the measured or calculated data are shown with box markers connected by a thin line. Bold lines show these data smoothed by a locally weighted function over 60 kyr.

#### 4.2 Strontium isotope dating used to support the new age model

The strontium isotope dating of the planktic foraminifera samples from Kottaphi Hill is in good agreement with the calcareous nannofossil dating. The ages of the samples produced by the new age model are within the error range calculated from the Sr isotope analysis.

In contrast, the Sr isotope dating of the samples from Lapatza Hill are problematic. Sedimentological study of the succession (Cannings, 2024) did not reveal any evidence of large-scale reworking or structural disturbance. Whilst diagenetic alteration is a possibility, this process did not produce

anomalous results in the stable isotope analyses of planktic foraminifera from the Lapatza Hill succession.

Previously anomalous <sup>87</sup>Sr/<sup>86</sup>Sr values have been documented for samples of Messinian age and have been attributed to the onset of the Messinian Salinity Crisis (MSC). For example, apparently anomalous <sup>87</sup>Sr/<sup>86</sup>Sr results have been documented from pre-evaporitic sequences in southern Turkey, and

- attributed to variations in <sup>87</sup>Sr/<sup>86</sup>Sr, due to isolation of small, relatively proximal basins prior to the Messinian Salinity Crisis (Flecker and Ellam, 1999; Flecker et al., 2002; Flecker and Ellam, 2006). The anomalous <sup>87</sup>Sr/<sup>86</sup>Sr values of all of the samples in this succession suggest that the onset of the MSC led to changes in the local <sup>87</sup>Sr/<sup>86</sup>Sr of seawater c. 2 million years before the deposition of the evaporites. Local changes in <sup>87</sup>Sr/<sup>86</sup>Sr are likely to represent changes in the balance between ocean input and local
- changes in the hydrological cycle, as suggested by previous modelling of <sup>87</sup>Sr/<sup>86</sup>Sr ratios in the Eastern Mediterranean basins (in SW Turkey) during the Late Miocene (Flecker et al., 2002). Modelling studies suggest that the closure of the eastern gateway of the southern proto-Mediterranean resulted in significant alterations to the hydrological cycle in the eastern Mediterranean region, with evaporation exceeding freshwater input (Karami et al., 2009), which in turn lead to variable <sup>87</sup>Sr/<sup>86</sup>Sr values. The
- most likely explanation for the lack of correlation of sample height in the succession with the calculated age in the Lapatza Hill succession is that the Sr isotope values of the planktic foraminifera analysed do not reflect the Sr isotope values of the global seawater at the time of biomineralization. Calcareous nannofossil biostratigraphy indicates that the conditions resulting in anomalous <sup>87</sup>Sr/<sup>86</sup>Sr values were present from c. 8 Ma until the end of the recorded data of this study.

# 525 4.3 Discrepancies between stable isotope records

As noted above, the  $\delta^{18}O_{\text{fine fraction}}$  record (Fig. 7) shows a dramatic offset towards much more negative  $\delta^{18}O$  values at the splice point between the Kottaphi Hill and Lapatza Hill successions. Similarly, the  $\delta^{13}C_{\text{fine fraction}}$  record shows a dramatic offset (c. 4.5 ‰) towards more negative  $\delta^{13}C$  values at the splice point. Following this offset, both fine fraction records show different trends from the planktic records

- for the same sample sets from Lapatza Hill.
  - Interpretations based on fine fraction measurements are generally not as well constrained as those based on planktic foraminiferal measurements, mainly because the former can be influenced by matrix composition. Facies observations, together with mineralogical and geochemical data, indicate that the samples from the Lapatza Hill succession have a greater terrigenous component compared to those from
- the Kottaphi Hill succession (Cannings, 2024). The isotopic signal of the terrigenous component was therefore necessarily measured together with the biogenic carbonate signal when the fine fraction bulk samples from the Lapatza Hill succession were analysed. Both the  $\delta^{18}O_{planktic}$  and the  $\delta^{13}C_{planktic}$  records do not show a dramatic offset at the splice point. This supports the interpretation that the offset in the fine fraction records is due to differences in the composition of the bulk samples.
- A relatively low magnitude shift is noted in the  $\delta^{13}C_{planktic}$  record compared to the fine fraction bulk records (Fig. 8). This shift is likely to represent small differences between the local  $\delta^{13}C_{seawater}$  at each of the two localities. Localised more negative  $\delta^{13}C$  values may indicate redeposition of isotopically light carbon from a nearby shelf during sea level changes, or the input of isotopically light carbon from the continent (Vincent et al., 1980; Vincent and Berger, 1985; Kouwenhoven et al., 1999).

For the Kottaphi Hill succession, the fine fraction records are mainly influenced by the biogenic carbonate material present, as evidenced by the similarity between the fine fraction and the planktic records. As a result, the fine fraction records for Lapatza Hill are not suitable for interpreting climatic or oceanographic changes. The planktic record for Lapatza Hill is, however, relatively complete, limiting any need for an interpretation based on the fine fraction record. On the other hand, the fine fraction measurements of samples from Kottaphi Hill appear to be controlled by the stable isotopic composition of biogenic calcite. Therefore, the Kottaphi Hill fine fraction records can aid interpretations of climatic and oceanographic changes. This is especially useful for the interval from *c*. 19 Ma to *c*. 17 Ma when planktic foraminifera are absent or poorly preserved.

#### 4.4 Palaeoceanographic implications

# 555 4.4.1 $\delta^{18}$ O

Some useful interpretations and regional to global comparisons concerning oxygen isotope trends, palaeotemperature and climatic evolution can be made based on the stable isotope records presented here (Fig. 10), in the light of the geological setting.

- The trend towards more negative  $\delta^{18}O_{\text{fine fraction}}$  values suggests that some warming occured during the deposition of the lowermost part of the Kottaphi Hill succession sampled between 18.96 Ma and 17.08 Ma. Relatively low  $\delta^{18}O_{\text{fine fraction}}$  and  $\delta^{18}O_{\text{planktic}}$  values from 17.08 Ma to 14.78 Ma can be correlated with the Miocene Climatic Optimum. This period of low  $\delta^{18}O$  was punctuated by three periods of inferred very high temperatures. The most prominent of these extreme periods took place at c. 15.7 Ma, c. 15.3 Ma and c. 14.9 Ma, consistent with an approximately 400 kyr cyclicity. Warm peaks on the c.
- 400-kyr eccentricity cycle are known from palaeotemperature records of the Pacific Ocean (Miller et al., 2005; Miller et al., 2020). These intervals are assumed to represent an ice-free Earth during the MCO, possibly the most recent ice-free periods in Earth history (Miller et al., 2005; Miller et al., 2020). The beginning of the δ<sup>18</sup>O increase recorded at *c*. 14.8 Ma correlates with Mi3a, marking the end of the MCO and the start of the MMCT (Miller et al., 1991a, b, 2020). Another positive δ<sup>18</sup>O excursion at 13.8
- 570 Ma marks the Mi3 event, representing the second cooling step associated with the MMCT (Miller et al., 1991a, b, 2020).
  - Open-ocean temperature records show slight warming and relatively stable temperatures following the first two cooling steps of the MMCT (Zachos et al., 2008; Miller et al., 2020; Westerhold et al., 2020). In contrast, the decreasing  $\delta^{18}O$  as recorded during this interval in Cyprus appears to indicate a local
- warming event. The cause of this apparent decoupling between the global climate and the eastern Mediterranean climate (using Cyprus as a reference) is uncertain but could relate to the dramatic restriction of the oceanic gateway between the proto-Mediterranean Sea and the Indian Ocean, which decreased the interchange between these two water masses. For example, the increasingly narrow shallow water connection could have increased the volume of relatively warm seawater (heated directly
- by insolation and by warm river water input) input to the eastern Mediterranean basin. Both geological evidence (Hüsing et al., 2009; Robertson et al., 2016; Torfstein and Steinberg, 2020) and geochemical evidence (Bialik et al., 2019; Torfstein and Steinberg, 2020) indicate that the collision of the Arabian and Tauride continental crust was well advanced by the Early Miocene (c. 20 Ma). This progressive closure left a mainly shallow seaway between the Indian Ocean and the Mediterranean Sea until

- complete closure at c.11 Ma (Hüsing et al., 2009). The shift to more negative  $\delta^{18}$ O values between 13 and 11 Ma in the records from Malta has also been attributed to the closure of this ocean gateway (Jacobs et al., 1996).
  - Following the above period of decreased  $\delta^{18}$ O, a positive excursion existed between 12.8 and 12.42 Ma, corresponding to the final cooling step of the MMCT and Mi4 (Miller et al., 1991b, 2020). The negative
- $\delta^{18}$ O excursion at *c*. 10.8 Ma was coeval with a warming event, as identified in a  $\delta^{18}$ O<sub>benthic</sub> record from the Western Pacific Ocean (Holbourn et al., 2013). This transient warming event is known as the Tortonian Thermal Maximum and is also reported in global temperature records (Westerhold et al., 2020). The warming event in the Western Pacific Ocean was transient and ended after <100kyr. However, the decreasing  $\delta^{18}$ O trend recorded in the eastern Mediterranean continued until *c*. 10.2 Ma.
- An approximately coeval warming interval between c. 11 Ma and c. 10 Ma is noted in Mg/Ca palaeotemperature records from the Equatorial Atlantic Ocean but this is not recorded in the Pacific Ocean (Lear et al., 2003). By this time period, temperature changes in the proto-Mediterranean appear to reflect those in the Atlantic Ocean rather than the Pacific Ocean. This is consistent with limited connectivity between the proto-Mediterranean and the Indian Ocean by this time.
- From c. 10.2 Ma, a long-term increasing  $\delta^{18}$ O trend continued until the end of the record. This apparent cooling trend aligns with the long-term cooling trend between c. 10 Ma and c. 6 Ma, as inferred from  $UK_{37}$ ' palaeotemperature records, and has been related to acceleration of Antarctic glaciation (Herbert et al., 2022). This positive  $\delta^{18}$ O trend was gradual until a more dramatic increase in  $\delta^{18}$ O at c. 6.8 Ma. This may correspond to intense cooling from c. 7 Ma and c. 6 Ma, termed 'Late Miocene Cooling'
- (LMC), which is recorded in  $UK_{37}$ ' palaeotemperature records for high, mid and tropical latitudes in both hemispheres of the Atlantic and Pacific oceans (Herbert et al., 2016; Tanner et al., 2020). Following the above dramatic  $\delta^{18}$ O increase, the recorded  $\delta^{18}$ O measurements rapidly increase, consistent with Late Miocene warming. However, this should be taken with caution because of the extensive changes in the hydrological cycle that are attributed to the Messinian Salinity Crisis (Flecker and Fllow 1999, 2006; Flocker et al., 2002). As a result, high salinity levels are likely to have been an
- and Ellam, 1999, 2006; Flecker *et al.*, 2002). As a result, high salinity levels are likely to have been an important control on  $\delta^{18}$ O in foraminiferal calcite during this interval.

#### 4.4.2 $\delta^{13}$ C

- The  $\delta^{13}$ C<sub>fine fraction</sub> record from Kottaphi Hill shows a period of elevated  $\delta^{13}$ C values between c. 17 Ma and c. 13.5 Ma (Fig. 8). This is also recorded in the  $\delta^{13}$ C<sub>planktic</sub> record, albeit not as well defined (Fig. 8).
- This interval correlates with the well-documented Monterey global carbon isotope event (Vincent and Berger, 1985; Holbourn et al., 2007; Holbourn et al., 2015) (Fig. 10). The Monterey Event is associated with increased biological carbon isotope fractionation under high CO<sub>2</sub> conditions, together with enhanced burial of organic matter on continental shelves presumably as a result of eustatic sea-level rise (Sosdian et al., 2020).
- Both the  $\delta^{13}C_{\text{fine fraction}}$  and the  $\delta^{13}C_{\text{planktic}}$  data for the composite temporal record show a gradual decrease in  $\delta^{13}C$  following the Monterey Event (Fig. 10). This decrease, which is also noted in open ocean records (Cramer et al., 2009), has been related to increased CO<sub>2</sub> drawdown following the high CO<sub>2</sub> conditions of the Monterey Event (Vincent and Berger, 1985; Flower and Kennett, 1993; Flower and Kennett, 1994; Sosdian et al., 2020). Both the  $\delta^{13}C_{\text{fine fraction}}$  and  $\delta^{13}C_{\text{planktic}}$  are especially low at c.

- 12 Ma. This trend towards more negative  $\delta^{13}C$  may also relate to reduced primary productivity in the ocean, in response to the onset of the Miocene Carbonate Crash (Holbourn et al., 2018; Holbourn et al., 2021).
  - The  $\delta^{13}$ C<sub>fine fraction</sub> and  $\delta^{13}$ C<sub>planktic</sub> records both show a shift to more positive values at c. 10.8 Ma (Fig. 10). This corresponds to the Tortonian Thermal Maximum, which is recorded at a high magnitude in the  $\delta^{18}$ O records (Fig. 7 & Fig. 10).
- δ<sup>18</sup>O records (Fig. 7 & Fig. 10).
  At 7.2 Ma, a dramatic shift (c. 2.5 ‰) to more negative carbon isotope values is shown in the δ<sup>13</sup>C<sub>planktic</sub> record (Fig. 10). This shift corresponds to the Late Miocene Carbon Isotope Shift (LMCIS) and the onset of the Late Miocene-Early Pliocene Biogenic Bloom (Dickens and Owen, 1999; Diester-Haass et al., 2004; Diester-Haass et al., 2005). The LMCIS has been linked to the intensification of global
- cooling and the Asian winter monsoon during the Late Miocene (Holbourn et al., 2018). The LCMIS has been explained by changes in ocean circulation, atmospheric circulation, or a combination of other factors such as redistribution of nutrients, changes in photosynthetic pathways or terrestrial weathering (Dickens and Owen, 1999; Diester-Haass et al., 2005; Holbourn et al., 2015; Tzanova et al., 2015; Herbert et al., 2016; Lyle et al., 2019). Although the LMCIS is recorded in the new δ¹³C<sub>planktic</sub> record,
- after this time the  $\delta^{13}$ C<sub>planktic</sub> record does not appear to reflect the open ocean record from the North Atlantic Ocean (Cramer et al., 2009). This suggests that after c. 7.2 Ma (early Messinian), the  $\delta^{13}$ C composition of the seawater in the eastern Mediterranean, specifically Cyprus, was no longer strongly influenced by global  $\delta^{13}$ C changes. A change in benthic foraminiferal assemblages and a shift to more negative  $\delta^{13}$ C values in the West Alboran Basin (Western Mediterranean) at 7.17 Ma has been related to
- restriction of the Mediterranean-Atlantic gateway (Bulian et al., 2022). In the Lapatza Hill succession, a change in the benthic foraminiferal assemblage is noted at 7.23 Ma. Distinguishing between the effects of the LMCIS and the restriction of gateways in the west and/or the east is difficult. However, some effect of gateway closure is suggested by the discrete  $\delta^{13}$ C trend recorded in the Lapatza Hill succession after c. 7.2 Ma. The well- documented constriction of the eastern gateway to the Indian Ocean during
- the Early Miocene, especially in SE Turkey (Hüsing et al., 2009; Bialik et al., 2019; Torfstein and Steinberg, 2020) is likely to have influenced the more negative carbon isotope values.

Figure 10 Plot of planktic and fine fraction δ13C and δ18O versus age for the composite temporal record. Measurements are shown relative to VDPB. Both smoothed and calculated data are shown, as in Fig. 9. The timings of the Mi2–Mi7 events (Miller et al., 1991a; Miller et al., 2020) are also shown. The timing of several global to regional-scale oceanographic events are indicated: (1) The Monterey Event (Vincent and Berger, 1985; Holbourn et al., 2015); (2) Miocene Carbonate Crash (Lyle et al., 1995; Lübbers et al., 2019); (3) Biogenic Bloom (Dickens and Owen, 1999; Diester-Haass et al., 2004; Diester-Haass et al., 2005); (4) Late Miocene Carbon Isotope Shift (LMCIS) (Hodell et al., 1994); (5) The Miocene Climatic Optimum (Zachos et al., 2008; Holbourn et al., 2015; Westerhold et al., 2020); (6) Restriction (from c. 20 Ma) and closure of the eastern gateway of the proto-Mediterranean (Hüsing et al., 2009; Bialik et al., 2019); (7) The Miocene Climate

Transition (Holbourn et al., 2015; Westerhold et al., 2020); (8) The Tortonian Thermal Maximum (Westerhold et al., 2020); (9) Late Miocene Cooling (Herbert et al., 2016); (10) The Messinian Salinity Crisis (Krijgsman et al., 1999a).

#### 4.5 Wider implications

This is the first-of-its-kind long-term δ<sup>13</sup>C and δ<sup>18</sup>O stable isotope record for the eastern Mediterranean basins. This new record provides a useful reference section for future studies of Miocene eastern Mediterranean palaeoceanography. Whilst the trends recorded for global events such as the MMCO, MMCT and the Monterey event were anticipated, this new record additionally reveals an extreme and long-lasting oxygen isotope excursion that coincides with the Tortonian Thermal Maximum. The apparent discrepancy in magnitude between this event in Cyprus and that recorded in global records warrants further consideration. The discrepancy could relate to the developing restriction in connectivity between the eastern Mediterranean Sea and the open ocean.

#### **5 Conclusions**

- A shift in  $\delta^{13}$ C and  $\delta^{18}$ O fine fraction values between two successions studied in Cyprus, the Kottaphi Hill succession and the Lapatza Hill succession, is explained by a larger terrigenous component in samples from the Lapatza Hill succession. Although fine-fraction samples can be used effectively for sites with a low terrigenous input (such as Kottaphi Hill), care must be taken to avoid recording signals unrelated to biogenic carbonate material where background terrigenous input is high.
- The Early-Middle Miocene Monterey Event is clearly recorded in the composite temporal  $\delta^{13}$ C record that was produced by splicing of the two correlative successions (Lapatza Hill and Kottaphi Hill).
- The new composite δ<sup>18</sup>O record from Cyprus allows the recognition of global climate events including the Miocene Climatic Optimum, the Middle Miocene Climate Transition and the Late Miocene Cooling. The decreasing δ<sup>18</sup>O trend, beginning at the Tortonian Thermal Maximum appears to have continued for longer in the eastern Mediterranean Sea (using Cyprus as a reference) compared to global records. This could represent a local to regional scale warming event during a time when there was apparently extensive variation and disequilibrium, both between and within the Atlantic and Pacific oceans.
- A negative shift in  $\delta^{13}$ C<sub>planktic</sub> values at c. 7.2 Ma may correlate with the global 'Late Miocene Carbon Isotope Shift'. However, it is difficult to distinguish the effects of the LMCIS from regional  $\delta^{13}$ C trends in Cyprus due to the partial closure of the Indian Ocean-Mediterranean and Mediterranean-Atlantic Ocean gateways.
- This new composite stable isotope record from Cyprus also provides a useful reference section for the study of oceanographic changes in the eastern Mediterranean basins during the Miocene, compared to the western Mediterranean basins and globally.

#### 695 Author Contribution

TC, AHFR and DK designed the research and carried out the initial fieldwork together. TC carried out the nannofossil dating and the analysis. TC, AHFR, and DK discussed and interpreted the resulting data. TC produced the figures. TC and AHFR wrote the manuscript.

# **Competing Interests**

The authors declare no conflict of interest.

#### Acknowledgements

This paper is based on studies carried out for the PhD degree by T.C. at the University of Edinburgh. This research was funded by a Natural Environment Research Council E<sup>4</sup> DTP studentship (NE/S007407/1). We thank Erica de Leau for her help and support with this research. We are also grateful to Dr. Simon Jung for his help with the production of the age model. We thank Colin Chilcott and Dr. Ulrike Baranowski for their assistance with stable isotope analysis. We also thank Anne Kelly, Vincent Gallagher and Prof. Darren Mark, Scottish Universities Environmental Research Centre, East Kilbride, for their assistance with the Sr isotopic analysis. T.C. thanks Dr. Mehmet Necdet, Dr. Elizabeth Balmer and Jacob Shearer for their help during fieldwork, and also Prof. Isabella Raffi for her guidance and training in calcareous nannofossil biostratigraphy. The authors also thank Prof. Isabella Raffi and Prof. Helmut Weissert for reviewing the article and providing valuable guidance for its improvement.

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
