# Peer review of "Insights into the Middle-Late Miocene palaeoceanographic development of Cyprus (E. Mediterranean) from a new $\delta^{18}O$ and $\delta^{13}C$ stable isotope composite record"

_EGUsphere, 2025_

## Author Response (AR1)

Response to Reviewers - Insights into the Middle-Late Miocene palaeoceanographic development of Cyprus (E. Mediterranean) from a new  $\delta^{18}$ O and  $\delta^{13}$ C stable isotope composite record

I would like to take this opportunity to once again thank both reviewers for their thoughtful evaluation of the article and for providing valuable guidance to improve it. The suggested changes and additions undoubtedly enhance the manuscript's clarity and coherence, and I am grateful for the reviewers' and editor's feedback.

In addition to the specific revisions outlined below, we have made further edits throughout the manuscript to improve its overall readability. Most of these are minor changes aimed at enhancing clarity and flow. Several of these edits also align with Reviewer 2's advice to shorten the introduction and reduce redundancy with the discussion section. We have also corrected and updated several references to ensure the accuracy of the citation details.

Best wishes,

**Torin Cannings**

**R1 -**

**Point -** A marginal observation regards the calcareous nannofossil biostratigraphy applied in this study: besides the used reference scheme (by Backman et al., 2012), that is appropriate for providing a reliable age model, the Authors could address a more recent biostratigraphic review on nannofossil biostratigraphy in the Mediterranean [the reference is: A. Di Stefano, N. Baldassini, I. Raffi, E. Fornaciari, A. Incarbona, A. Negri, S. Bonomo, G. Villa, E. Di Stefano, D. Rio (2024). Neogene-Quaternary Mediterranean Calcareous Nannofossil Biozonation and Biochronology: a Review. Stratigraphy, 20(4): 259-302. https://doi.org/10.29041/strat.20.4.02]. Anyway, it is not a mandatory suggestion because it will not change the biochronologic results presented.

**Response** – A sentence has been added to the methods section (2.4) stating that results are consistent with more up to date scheme.

**R2 -**

**Point** - I recommend that the authors shift at least part of the long climate and ocean introduction (around lines 90-125, possibly also some of sentences in lines 55-60) to the discussion. This minimizes redundancies in the discussion.

**Response -** Several sections of the introduction have now been moved to the discussion to remove redundancy. Some parts of these sections were left in order to preserve the flow of the introduction and ensure that all key events are properly introduced. This is important as we hope to interest land-based geologists who may not be up to date with regional to global palaeoceanography. On the other hand, many palaeoceanographers may not be familiar with the sedimentary geology of Cyprus. Where possible the introduction has been shortened and reworded for clarity.

**Point -** Additional references in the introduction: Cedric John et al., 2003, GSA, on Miocene climate in the Mediterranean.

**Response –** This reference has been added to the introduction (1.1).

**Point -** Line 117: see also impact of closure of gateway to the east on Mediterranean isotope records in Jacobs et al., 1996.

**Response** – This reference has been added to the discussion (4.4.1) (part of introduction moved to discussion) and removed from the introduction (2.1.1) where originally mentioned.

**Point -** 310 are condensed intervals comparable to phosphorites in the central and western Mediterranean? (see Jacobs et al., hardgrounds 16.9 and 16. 1 Ma). Any evidence for changing deep current intensities?

**Response -** The 'firmground' interval at Kottahi Hill may be comparable to those observed in Malta. While a detailed analysis of these layers was beyond the scope of this study, a future effort to compare and potentially correlate such intervals across the Mediterranean would be a valuable direction for further research. As these intervals were not studied in detail as part of this work, a detailed comparison with layers elsewhere cannot be made.

**Point -** 415 onwards: stable isotope values expressed in the d-notation are not "heavier" or "lighter" but more positive/negative.

Response - This phrasing has now been changed in all appearances in the text.

Point - 484 correct the title

**Response –** This typo has been corrected.

**Point -** From 499 onwards: Please add isotope events (Mi 3, 3a, 4, MMCT, LMCIS etc.) to your summary figure 10. This will facilitate readers comprehension of your argumentation.

Response - Isotope events have now been added to the figure and caption alongside key events.

Point - 519 delete "n"

Response - This has been edited to say 'in", this was a typo where the 'i' from "in" was missing.

**Point -** 525-530 > some duplication, see your introduction, lines 65-75

**Response** – The duplication has been removed as part of the changes made to move parts of the introduction to the discussion. We have shortened and streamlined the text by careful re-editing. Any unnecessary details or minor repetitions were removed.

---

## Author Response (AR2)

Dear Authors,

As indicated in the public response, the text reads now fine. There are, however, issues with some of the figures. This is aside from most figures being rather fuzzy, which I trust will be corrected in the production phase of the manuscript. The first issue relates to the first occurrence indication in figure 4 of Amaurolithus primus (Lapatza Hill section). In the text this first occurrence is reported at c. 18.5m (line 390). In the figure, however, the arrow indicates a depth of 19.5m. Could this please be looked into and other, comparable references be checked? This might involve minor text edits as well.

The second issue affects figures 7-10, more specifically the stable O and C isotope records of the fine fraction. The is a mismatch between the smoothed version of the records and the underpinning original records. In figure 8 for example, between roughly 8 and 6 Ma, the smoothed and the original d13C records are clearly offset. This likely explains why the original (unsmoothed) records end earlier than the smoothed versions (does not really make sense). Smaller issues are indicated in the carbon isotope record between roughly 12 and 8.6 Ma. Can these issues please be looked into as well?

If there are questions, please let me know.

Best wishes,

Simon

Dear Simon,

Thank you very much for your message and for bringing these issues to our attention. They have now been rectified as detailed in the text below.

Best wishes,

Torin

**All Figures** – This is aside from most figures being rather fuzzy, which I trust will be corrected in the production phase of the manuscript.

This was due to compression during conversion of the manuscript to PDF. All figures will be submitted as high-resolution image files in the final version.

**Figure 4** – In the text, this first occurrence is reported at c. 18.5m (line 390). In the figure, however, the arrow indicates a depth of 19.5m.

This was a typo in the text, which has now been corrected. Thank you for bringing it to our attention. This discrepancy did not affect the calculated sedimentation rates or any subsequent text.

**Figures 7–10** – The stable O and C isotope records of the fine fraction. There is a mismatch between the smoothed version of the records and the underpinning original records.

Thank you very much for pointing this out — I should have caught it myself. The issue stemmed from the original (unsmoothed) record being plotted on an outdated age model. This has now been corrected, and all relevant figures have been updated accordingly.